# Modeling of a Digital Twin for Magnetic Bearings

**Omer W. Taha** [1,2,*] and **Yefa Hu** [1]

1 School of Mechanical and Electronic Engineering, Wuhan University of Technology, Wuhan 430070, China
2 Computer Technology Engineering, Al-Mustafa University College, Baghdad 10067, Iraq
* Correspondence: omer.huish@outlook.com

**Abstract:** As an essential enabling technology to realize advanced concepts such as digitization, intelligence, and service, information technology plays a critical role in shaping modern society and driving innovation across various industries and domains. The concept of the digital twin is attracting attention from academics and industry, and how to apply it in various fields. In this paper, the performance of the magnetic bearing system may be simulated in real-time using a digital twin, especially the resulting vibration from the unbalanced rotor mass, which caused a drop in performance and a high risk of system instability and potential safety accidents. It is suggested to use a model-data combination driven digital twin model to examine its dynamic characteristics and vibration mechanism. The vibration data of the magnetic bearing was collected through experiments and compared with the data derived from the simulation results. The efficiency of the suggested strategy is demonstrated by confirming that digitally anticipated vibration signals are consistent with physical space measurements. The result shows that the fine digital twin geometric model of magnetic bearing is more consistent with the actual operation. By allowing the identification of problems before they become critical, using a digital twin may increase the dependability of magnetic bearings while reducing the possibility of unexpected downtime or failures.

**Keywords:** digital twin; magnetic bearing; vibration; mass imbalance





## 1. Introduction

Magnetic bearings are a relatively new technology that has been gaining popularity recently, particularly in industrial machinery and equipment [1]. Unlike conventional bearings that rely on physical contact between the rotor and the stator, magnetic bearings use electromagnetic forces to levitate and stabilize the rotor [2]. This allows for several advantages over standard bearings, such as increased efficiency, excellent reliability, and the ability to operate in harsh environments [2]. However, these benefits come at the cost of increased complexity and the need for precise control and monitoring of magnetic forces [3]. A digital twin is a virtual representation of a physical asset or system that monitors, analyzes, and optimizes performance [4]. A digital twin may be utilized to simulate the behavior of a magnetic bearing system in real time, giving important insights into the system's performance and detecting possible problems before they develop. The control algorithms used to steady the rotor may also be enhanced with the help of the digital twin, leading to better performance and less energy usage.

Building a digital twin of a magnetic bearing system requires accurate and detailed models of the system and real-time sensor data from the physical system [5,6]. The digital twin must be able to accurately simulate the physical behavior of the system, including the electromagnetic forces that are used to levitate the rotor [7]. Additionally, the digital twin must be able to incorporate real-time sensor data from the physical system, such as the position and speed of the rotor, to provide accurate and up-to-date information on the system's performance [8].

One of the key challenges in building a digital twin of a magnetic bearing system is the need for precise control and monitoring of the magnetic forces [9]. This requires

high-precision sensors and control algorithms that can accurately measure and control the magnetic forces, even in the presence of noise and other disturbances [10]. Additionally, the digital twin must be able to handle the complex dynamics of the system, including the interactions between the rotor and the stator and the effects of external disturbances such as vibrations and temperature changes [11,12]. Fei Tao et al. [13] realized the accurate detection of equipment failure by establishing an accurate digital model of equipment failure. Qianzhe Qiao et al. [14] modeled the equipment loss process and the normal operation state of the equipment to realize a condition assessment and equipment failure prediction. Yiwei Cheng et al. [15] used neural network technology and wavelet transform technology. The technique was used to analyze the equipment operation data, and the method's feasibility was verified by predicting the failure of rotating machinery. Qingli Dong et al. [16] discussed the fault diagnosis of wind turbines and proposed a method, based on the nonlinear model of the fuzzy system and neural network, to predict equipment failure and used this method to predict the wind. The failure of the power turbine was tested. Zhang et al. [17] used digital twin technology to achieve rapid design and optimization of glass production according to production needs, based on the actual glass combined with the glass production line. R. Mudassar et al. studied how project management is shifting toward smart manufacturing in the present and the future; as a consequence, companies that are smart can take advantage of enhanced flexibility and quick lead times, as well as improved maintenance, product quality, resource usage, and efficiency monitoring [18]. To make wise planning choices to deal with the rising demand and profit-enhancing methods, M. Jabir et al. integrate information sharing from planning issues, printed circuit board (PCB) assembly lines, retailers, and shipping departments with a cloud manufacturing system (CMS) [19]. Mengjun Liu et al. suggested multi-level planning and scheduling integrated with material restrictions in rolling horizon intervals, and they used the theory of constraints and the Drum Buffer Rope (DBR) mechanism for multi-level planning and scheduling in mixed-model production. They also proposed a multi-level planning heuristic (MLPH) using the DBR and priority rules for efficient planning and scheduling, taking into account material requirement planning, and they used the capacity constraint resource (CCR). The suggested multi-level planning heuristic outperforms the fundamental scheduling principles such as EDD, SPT, and LPT in the outcomes [20]. Fei Tao et al. [21] actually applied digital twin technology to production machine tools. Sensors collect data related to production machine tools in the physical space and realize the fusion of actual production machine tools and virtual machine tools information. In order to further expand the scope of the application of digital twin technology, the 2012 American National Standards and Technology Research Academy defines two major categories: MBD (Model-Based Definition) and MBE (Model-Based Enterprise). This concept was used in [22], thereby extending the scope of application of digital twin technology to the entire product life cycle, including pre-production product design simulation, mid-term product processing and production, and after-sales maintenance of later products. Fei Tao et al. [23] combined web services technology and augmented reality technology to simulate the virtual body in digital twin technology. The visualization was explored.

Despite these challenges, the benefits of using a digital twin of a magnetic bearing system are clear. By providing real-time insights into the system's performance, the digital twin can help identify vibration for imbalance mass potential problems before they occur and optimize the control algorithms to improve performance and reduce energy consumption [24,25]. Additionally, the digital twin can simulate different operating conditions and test different control strategies, providing valuable information for designing and developing new magnetic bearing systems [26].

In conclusion, the digital twin is a powerful tool for monitoring, analyzing, and optimizing the performance of magnetic bearing systems. It provides real-time insights into the system's performance and helps identify potential problems before they occur. Additionally, it can be used to optimize the control algorithms and test different operating conditions, providing valuable information for the design and development of new magnetic bearing

systems. Despite the challenges, the benefits of using a magnetic-bearing digital twin are clear. It is expected to become a key technology in the field of industrial machinery and equipment. The objective of developing and modeling a digital twin for magnetic bearings is to create a virtual representation of the physical magnetic bearing system that can be used to optimize its performance, enable predictive maintenance, and diagnose faults. In this paper, the first section is an introduction, and the second section discusses the material and method used in modeling: the digital twin, magnetic bearing, and the digital twin modeling. The third section reports on the results obtained from the modeling.

## 2. Materials and Methods

### 2.1. Digital Twin

The digital twin is one of the promising digital technologies now under development to support digital transformation and decision making in multiple industries [27]. It is essentially a digital replica of a physical entity created by combining data from various sources such as sensors, software, and simulations [28]. Digital twins are often used in manufacturing, engineering, and construction industries to create a virtual model of a physical model or system [29]. A digital twin can test different scenarios, predict how the physical system will behave under various conditions, and optimize its performance without needing physical prototypes [30]. Moreover, digital twins can be used to monitor and analyze the performance of physical assets in real-time, enabling predictive maintenance, and improving efficiency [31]. By continuously monitoring and analyzing the data, the digital twin can provide insights into potential issues, allowing preventative measures to be taken before any significant problems arise [32].

Overall, digital twins are a powerful tool for improving the design, performance, and maintenance of physical assets, enabling greater efficiency, cost savings, and innovation. A simulation digital twin uses computer simulations to represent a physical system [33]. It involves modeling the physical system in a digital environment and simulating the system's behavior under various conditions [34]. Digital twin simulations are often used in manufacturing to evaluate and improve the performance of a physical system before it is built. Simulating the system's behavior can test different design configurations, predict how the system will behave under different conditions, and optimize its performance. This can save time and money by avoiding the need for physical prototypes [35,36]. Digital twin simulations may also be used to monitor and assess a physical system's efficiency in real-time. The system's performance, ability to predict future problems, and ability to take preventative action before difficulties occur can be found by merging real-time data from sensors and other sources with the simulation model [37,38]. Integrating and fusing data is a critical aspect of creating a digital twin. The virtual model is directed by the information and behavior generated by the physical model, and this can generate a substantial amount of data. Several data processing techniques, such as data mining, monitoring, fusion, and optimization may be necessary to manage and analyze these data effectively. By following this method, it is possible to guarantee that the virtual representation accurately and constantly reflects the physical one [39,40].

### 2.2. Electromagnetic Analysis

When designing and manufacturing a magnetic bearing, choosing the right number of poles is essential. The number of poles was chosen to be eight after the design was optimized to improve performance, as shown in Figure 1. All the dimensions are listed in detail in Table 1. The upper electromagnets in the y direction (vertical) obtained 5 A, while the electromagnets on the left and right sides obtained 3.3 A in the x direction (horizontal). This was carried out to determine the maximum holding force. The holding force is heavily dependent on three major factors. The first factor is the size of the air gap between the legs and the shaft; the bigger the gap, the more the force needs to be given. The depth of the apparatus in the z direction is the second factor; double the depth equals double the force. Finally, the thickness of the crowns affects how much force may be generated [41].

Finding the highest holding force in the y direction was the biggest issue. The radial AMB generates the x-axis force and the y-axis force, two perpendicular radial forces. The current in each of the four electromagnets is independently regulated as the four electromagnets operate in differential mode. Figure 1 depicts an eight-pole radial magnetic bearing.

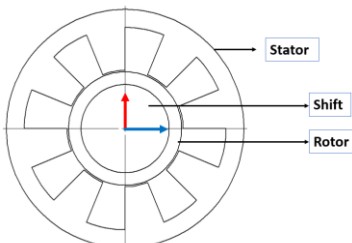

**Figure 1.** Radial active magnetic bearing.

**Table 1.** Different parameters used.

| Parameter | Value |
|---|---|
| Height of the stack | 20 mm |
| Shaft diameter | 24 mm |
| Operating air gap | 1 mm (radial) |
| The outer diameter of the MB | 100 mm |
| The inner diameter of the MB | 31 mm |

Figure 2 depicts an electromagnet used to suspend a C-shaped core using magnetic forces. The dotted line denotes the major flux path. Figure 2 shows how the C-core's flux route lengths are defined. The winding has N turns. When set to its default state, the air gap has a length g. The I-shaped core position's coordinate location x is established in order for the air gap length to be (g − x). The reluctance of the magnetic circuit is defined as

$$R = \frac{lf\mathrm{p}}{\mu mt\, S} \tag{1}$$

where $l_{f\mathrm{p}}$ = flux path length = $2D_i$, $\mu mt$ = permeability in the material, and S = cross-section area of flux path.

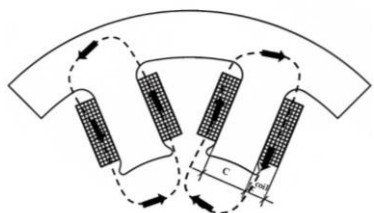

**Figure 2.** C-shaped cores.

Figure 3 depicts a comparable circuit for the electromagnet's magnetic circuit. A constant (dc) magnetic circuit may be thought of in terms of MMF (voltage), flux (current), and reluctance (resistance), much like an electric circuit. The primary distinction is that magnetic reluctance, the "dc voltage" produced by the winding current, is an energy storage component as opposed to a loss component. Rg stands for the magnetic reluctance at the air gap, while Rc is the magnetic reluctance [42]. These magnetic reluctances are written as

$$R_g = \frac{g - x}{\mu 0 \mu r \mathrm{W}} \tag{2}$$

$$R_c = \frac{2D_i + l}{\mu 0 \mu rW} \tag{3}$$

where

$\mu_0$ is the permeability of free space ($\mu_0 = 4\pi \times 10^{-7}$ H/m).

$\mu_r$ is the relative permeability ($\mu_{mt} = \mu r \mu 0$).

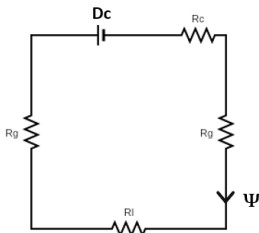

**Figure 3.** Equivalent electric circuit.

The relative permeability of air is approximately equal to 1.0.

The value of $\mu r$ for iron is typically in the range of 1000–10,000.

The magnetic reluctance in the iron may be disregarded in the following calculation since, in the majority of situations, the air gap reluctance is substantially more important than the iron reluctance. The flux is thus explained below, along with a simplified illustration of the comparable electrical circuit:

$$\Psi = \frac{N_i}{2R_g} = \frac{N_i}{2} \frac{\mu 0W}{g - x} \tag{4}$$

$$\text{flux linkage}(\lambda 1) = \frac{N^2_i}{2} \frac{\mu 0W}{g - x} \tag{5}$$

Inductance is defined as the flux linkage divided by the current value (L = l/i), while flux linkage is defined as the number of turns N multiplied by the flux going through the coil, giving

$$L = \frac{N^2}{2} \frac{\mu 0W}{g - x} \tag{6}$$

Nominal inductance $L_0$ is defined as

$$L_0 = \frac{N^2_i}{2} \frac{\mu 0W}{g} \tag{7}$$

In addition, the flux density $B_0$ in the air gap can be derived as Equation (8),

$$B_0 = \frac{\Psi}{W} = \frac{(g - x)}{N\mu 0 i} \tag{8}$$

Finally, after studying the equations of electric magnetic, the different parameters for a magnetic bearing (MB) are presented in Table 1:

Magnetic bearing simulation in solid work shown in Figure 4 depicts the current design of the AMB's layout. The shaft is in the core, and the four electromagnets are arranged around it. The legs are designed to provide a stronger grip by being rounded on the inside (crown style legs), with crown-shaped support legs at the pole.

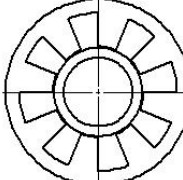 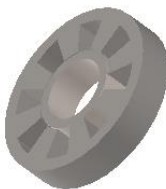

**Figure 4.** 3D model of magnetic bearing.

It is important to mention some definitions related to magnetic bearings.

A magnetic circuit that conducts the magnetic fields generated by an active magnetic bearing is called the stator. Thus, careful consideration should be given to choosing a suitable material for the stator. Understanding the properties of ferromagnetic materials is crucial when choosing a material for the stator. We chose a ferromagnetic material (silicon steel) with a high permeability and saturation level based on these properties. Sheet steel is the optimal stator material for our purposes because of its high permeability and low saturation levels. Laminations of fully laminated poles feature rivet holes for assembly and are made of sheet steel that is 0.5 mm thick. The laminations are constructed on steel rods and pushed between heavy steel plates at both ends. In an active magnetic bearing, a circular copper wire is used to wind the actuator coils. Enameller copper is utilized as insulation for these coils. Figure 5 present the actuator coils of an MB.

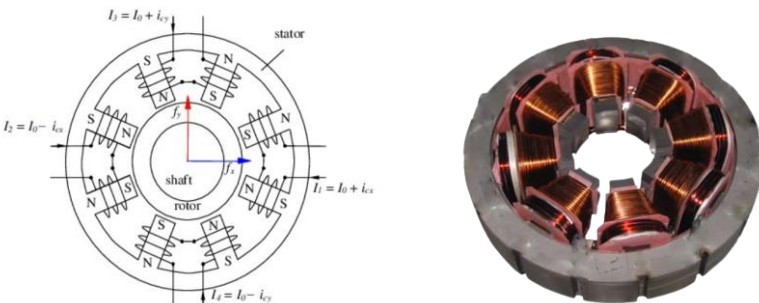

**Figure 5.** Actuator coils of magnetic bearings.

The rotor is a cylinder made of ferromagnetic material, such as laminated silicon steel, around a central shaft. The stator, which has eight poles, encloses the rotor. The windings are located in the slots between the stator poles. The magnetic paths of the eight stator poles. The stator's width is calculated to be sufficiently large so as to prevent magnetic saturation and provide high mechanical stiffness, thus reducing vibration due to radial magnetic forces. Due to the influence of the stator and rotor materials, the material will have magnetic saturation and hysteresis limitations. Whether the magnetic bearing can stabilize the suspension is the result of multiparty cooperation. The electromagnetic force differs from the design; hence, accurate data cannot be acquired with the present data acquisition methods. Still, the relevant functional connection may be roughly inferred by observing changes in the magnetic field, coil current, and rotor movement. It is a virtual entity that can approximately match the essential technical points of physical entities. The magnetic field analysis between the stator and rotor of the radial magnetic bearing is shown in Figure 6.

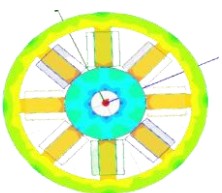

**Figure 6.** Magnetic field analysis between stator and rotor.

The primary considerations for data analysis of the magnetic bearing model in digital twins are the accuracy of real-time data collection, the amount of time required for analysis, how to handle the lag between feedback and analysis data, and related data processing. These factors all relate to the structure's force and vibration analysis, the effects of the magnetic field on the system, and the attempt to fit the approximate function relationship.

### 2.3. Modling of Digital Twin for Magnetic Bearing (MB)

With the introduction and widespread adoption of digital twins technology in recent years, it is now possible to bring the digital and physical worlds together in various applications [43]. For digital twin technology to be effective, it is crucial that the digital twin model precisely reflect the status of the actual work [44]. In the future, the user will receive both accurate and identical digital twin results. The digital twin modeling of magnetic bearings contains the dynamic, physical, and virtual layers. The magnetic bearing frame based on a digital twin is shown in Figure 7. The dynamic and static models are created based on the AMB's operating states, and the parameter update method for the digital twin model is obtained by collecting the physical entity's state data and combining them with the control algorithm, which enables the dynamic model to synchronously self-evolve with the static model. The operational state of the physical entity will be predicted using the synchronization parameters, which are then utilized to optimize the system control parameters for the realization of the composite control strategy driven by the digital twin. The virtual model describes the SolidWorks implementation. The dynamic model describes the closed-loop system for stable suspension and is established and controlled by a PID controller and fixed sensor on the magnetic bearing to obtain the virtual result. It is essential to mention that the ANSYS program used the geometric model to analyze the electric magnetic field for virtual data, as seen in the result of the magnetic bearing in Figure 6. The imbalance in the rotor's mass is then detected online, and the disk of balances is applied with equivalent counterweights to create a model of the rotor's vibrational dynamics. The model-data combined driven vibration digital twin model of a magnetic bearing is constructed. Finally, a comparison study is carried out between the results of the physical model and the results of the virtual model.

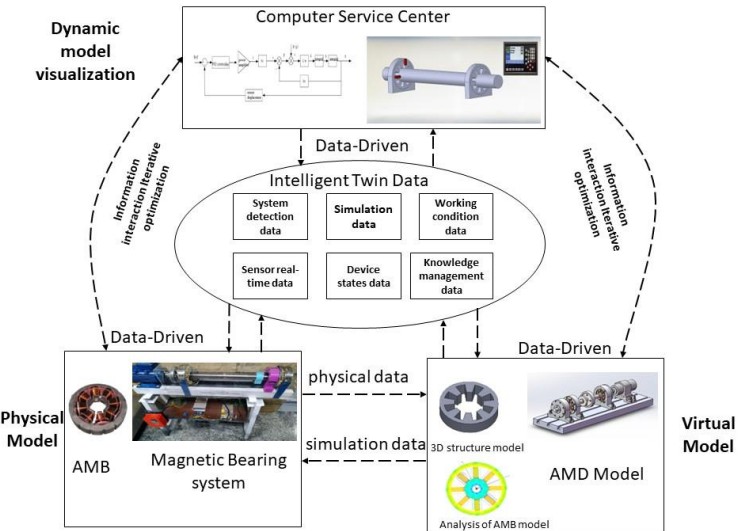

**Figure 7.** The design of the digital twin model for a magnetic bearing.

The geometric model of the solid shaft is depicted in Figure 8; it characterizes geometric properties such as the geometric parameters and assembly relationships explained below. The shaft has an iron core, a copper layer, a front spindle, a rear spindle, a nut, a counterweight disc, and a rotor for a magnetic bearing. The following diagrams and assemblies were created in SolidWorks. In particular, the relationship between the assembly

components is written as a multi-coordinate transform between the generalized coordinate system, the bearing coordinate system, and the sensor coordinate system.

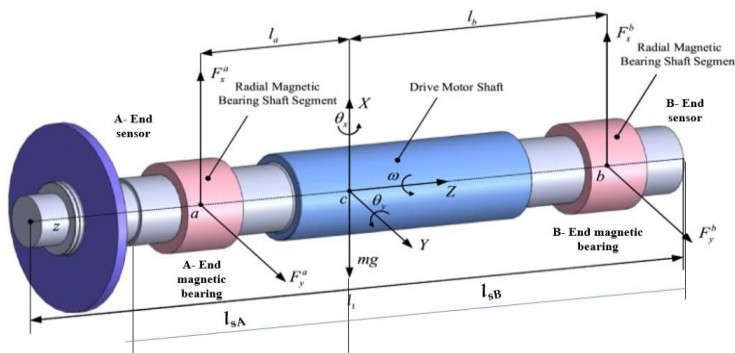

**Figure 8.** Geometric model of the magnetic bearing rigid shaft.

Here is a description of the coordinate systems involved. A generic coordinate system, CG-xyz, is developed with the geometric center of the shaft as its origin. The radial displacements are denoted by x and y, whereas the axial displacement is denoted by z. The angular velocity of a rotating shaft in the radial direction is denoted by ($\Omega$), qg = [$\beta$, x, $-\alpha$, y]$^T$ describes the movement in the generalized coordinate system. It is established that the bearing coordinate system has its genesis at the bearing center C.B. The $l_a$ and $l_b$ values represent the distances from the shaft geometry center to the A and B bearing centers. Bearing coordinate motion is denoted by the formula $q_b$ = [$x_a$, $x_b$, $y_a$, $y_b$]$^T$. The coordinate system of the sensors is set up with the C.S. at the sensor hub as the origin. The A-end and B-end sensors have a distance, $l_{sA}$ and $l_{sB}$, from the shaft geometrical center, qs = [$x_{sA}$, $x_{sB}$, $y_{sA}$, $y_{sB}$]$^T$ defines the motion in the sensor coordinate system. Six degrees of freedom (DOFs) are at play in the shaft's motion, including translation and rotation in the radial horizontal, radial vertical, and axial directions, respectively. Two axial–radial magnetic bearings bias provide suspension for the four-degrees-of-freedom radial motion. The radial dynamics (4-DOF) rotor system can be characterized as follows:

$$\left.\begin{array}{r} I_y\ddot{\beta} - I_z\Omega\dot{\alpha} = l_{bA}f_{xA} - l_{bB}f_{xB} \\ m\ddot{x} = f_{xA} + f_{xB} \\ I_x\ddot{\alpha} + I_z\,\Omega\dot{\beta} = l_{bB}f_{yB} - l_{bA}f_{yA} \\ m\ddot{y} = f_{yA} + f_{yb} \end{array}\right\} \tag{9}$$

The mass of the rotor is denoted by *m*. $I_x$, $I_y$, and $I_z$ indicate moments of inertia around the x, y, and z axes. Equation (10) presents $F_b$ = [$f_{xA}, f_{xB}, f_{yA}, f_{yB}$]$^T$. The bearing coordinate system represents the radial suspension forces acting on the four degrees of freedom.

$$Fb = K_x q_b + K_i I \tag{10}$$

*I = diag($i_{xA}$, $i_{xB}$, $i_{yA}$, $i_{yB}$)* denoted the current coil matrix of magnetic bearing.

$K_x = diag(K_{xA}, K_{xB}, K_{yA}, K_{yB})$ is the displacement stiffness matrix.

$K_i = diag(K_{iA1}, K_{iB1}, K_{iA2}, K_{iB2})$ is the current stiffness matrix, present in Equations (16)–(19).

The limitations for a magnetically suspended shaft can be summed up as 4-DOFs radial suspension forces $f_{xA}, f_{xB}, f_{yA}$, and $f_{yB}$, according to the mathematical model of rotor dynamics described by Equation (10).

The dynamic equation of the x plane is represented by the upper part of [M], [K], and [G], while the lower part represents the dynamic equation of the y plane. They are all 4N x4N-order sparse symmetric matrices, as shown in Equation (11).

The dynamic equations present below at the non-rotor ends and non-supported:

$$M_i\ddot{U}_i + G_i\dot{U}_i + K_i^1 U_{i-1} + K_i^2\,U_i + K_i^3\,U_i + K_i^4\,U_{i+1} = F \tag{11}$$

Substituting Equation (11) into Equation (10), to be expressed the dynamics of rotor system:

$$m\ddot{q}_g + G\dot{q}_g = T_b K_x T_b^T q_g + T_b K_i I \tag{12}$$

M represents the mass matrix, G is the gyro matrix, and Tb is the bearing coordinate system transformation matrix. Matrices can be expressed as below:

$$M = \begin{bmatrix} I_y & 0 & 0 & 0 \\ 0 & m & 0 & 0 \\ 0 & 0 & -I_x & 0 \\ 0 & 0 & 0 & m \end{bmatrix} \tag{13}$$

$$G = \Omega \begin{bmatrix} 0 & 0 & I_z & 0 \\ 0 & 0 & 0 & 0 \\ -I_z & 0 & 0 & 0 \\ 0 & 0 & 0 & 0 \end{bmatrix} \tag{14}$$

$$T_b = \begin{bmatrix} I_{bA} & -I_{bB} & 0 & 0 \\ 1 & 1 & 0 & 0 \\ 0 & 0 & I_{bA} & -I_{bB} \\ 0 & 0 & 1 & 1 \end{bmatrix} \tag{15}$$

The current stiffness matrix ($k_i$) can be written as the equations below, where Jz stands for the polar moment of inertia of the z-axis. I is the section inertia moment of the shaft segment, and I = $\frac{\pi d^4}{64}$

$$K_{iA1} = \begin{matrix} -12EI & 0 & -6EI & 0 \\ 0 & -12EI & 0 & -6EI \\ 6EI & 0 & 2EI & 0 \\ 0 & 6EI & 0 & 2EI \end{matrix} \tag{16}$$

$$K_{iA2} = \begin{matrix} 12EI & 0 & 6EI & 0 \\ 0 & 12EI & 0 & 6EI \\ -6EI & 0 & -2EI & 0 \\ 0 & -6EI & 0 & -2EI \end{matrix} \tag{17}$$

$$K_{iB1} = \begin{matrix} 12EI & 0 & 6EI & 0 \\ 0 & 12EI & 0 & -6EI \\ 6EI & 0 & 4EI & 0 \\ 0 & 6EI & 0 & 4EI \end{matrix} \tag{18}$$

$$K_{iB2} = \begin{matrix} -12EI & 0 & 6EI & 0 \\ 0 & -12EI & 0 & 6EI \\ -6EI & 0 & 2EI & 0 \\ 0 & -6EI & 0 & 2EI \end{matrix} \tag{19}$$

The physical layer specifies the virtual work design of the magnetic bearing, especially the rotor system's physical characteristics, including material parameters and limitations. In a radial magnetic bearing, there are two current amplifiers, each one of two pole pairs of electromagnets. By adjusting the electromagnetic forces of the bearing actuators in response to the rotor's position, the position controller keeps the rotor balanced at center C. Two main material parameters are affected: Poisson's ratio and Young's modulus. For the closed-loop rotor system, the specific co-simulation stages are shown in Figure 9. About the position regulator of the rotor, each bearing axis is individually controlled by a PID controller. Figure 8 presents the block diagram of the PID controller.

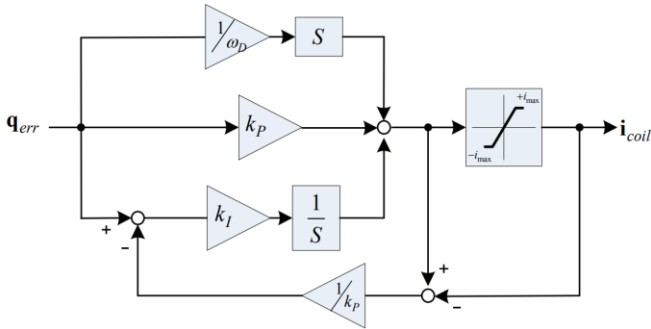

**Figure 9.** Block diagram for PID controller for each axis of the rotor.

Table 2 displays the dimensions and parameters of a rigid rotor model supported by radial active magnetic bearings, whereas Table 3 displays the control parameters of a local active magnetic bearing actuator.

**Table 2.** The dimensions and parameters of a rigid rotor model.

| Symbol | Meaning | Value | Units |
|---|---|---|---|
| $m$ | Mass of rotor | 24.26 | kg |
| $I_z$ | Inertia of rotor in z axis | 0.04525 | kgm$^2$ |
| $I_x = I_y$ | Inertia of the rotor in x, and y axis | 0.67388 | kgm$^2$ |
| $a$ | Position of Bearing A | $-0.2607$ | m |
| $c$ | Sensor position of bearing A | $-0.1107$ | m |
| $ksA$ | Position stiffness of bearing A | $-3.4 \times 10^6$ | N/m |
| $KiA$ | Current stiffness of bearing A | 765 | N/A |
| $b$ | Position of Bearing B | 0.2883 | m |
| $d$ | Sensor position of bearing B | 0.1373 | m |
| $ksB$ | Position stiffness of bearing B | $-3.4 \times 10^6$ | N/m |
| $KiB$ | Current stiffness of bearing B | 765 | N/A |

**Table 3.** The control parameters of a local active magnetic bearing actuator.

| Symbol | Meaning | Value | Units |
|---|---|---|---|
| $k_P$ | Proportional control gain | 0.4 | A/V |
| $k_s$ | Sensor gain | 0.02 | V/μm |
| $k_I$ | Integral control gain | 0.1 | A/V |
| $kPi$ | Proportional control gain | 45 | V/A |
| $Vdc$ | DC link voltage | 124 | V |
| $imax$ | Bias current of a magnet coil | 1.4 | A |
| $\omega_D$ | Derivative control gain | 243 | rad/s |
| $L$ | Inductance of a magnet coil | 124 | mH |
| $R$ | Resistance of a magnet coil | 0.453 | ohm |

The electromagnet is driven by a constant voltage from the dc-link, maintained by a simple proportional current controller acting as a current amplifier model, as seen in Figure 10. We also take into account the voltage's saturation level. The dynamics of the coil were taken into account. The current amplifier has a fixed output voltage of Vdc, where

Vdc is the amplifier's D.C. input voltage. Each electromagnet's behavior follows a linear relationship with control current and position variations.

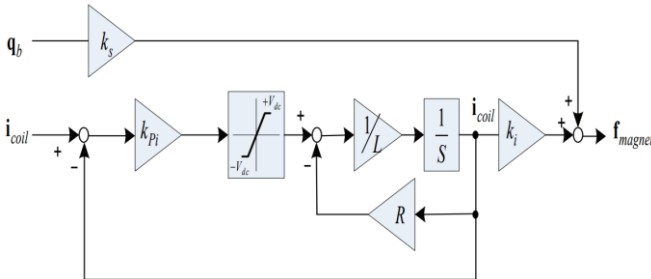

**Figure 10.** Current amplifier model.

Now, the important layer of digital twin modeling is explained briefly, which is named the behavior layer. Figure 11 shows that the behavior layer contains three modes.

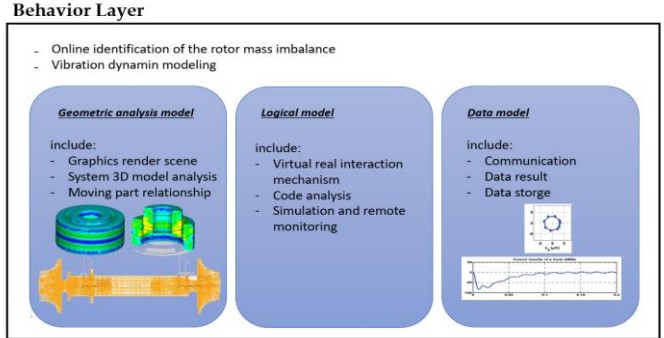

**Figure 11.** Behavior layer.

The first mode is the geometric analysis model, the second is the logical model, and the third is the data model to monitor the work of the vibration rotor with mass imbalance.

Generally, in the behavior layer, we identify the rotor mass imbalance and dynamic vibration modeling online based on the three modes of this layer.

The geometry analysis model includes the independent variables of rotational speed into the dynamics model and extends it to three-dimensional (3D) space obtained by the Ansys Twin Builder program. Furthermore, the parameter of the physical model for the rotors system (A and B) are obtained from sensor A and sensor B. The sensors' parameters are transferred to the behavior layer using MATLAB Simulink's closed-loop suspension control design with Ansys Twin Builder 3D dynamics modeling.

First, the assembled 3D geometry from Figure 12 is imported into Ansys Twin Builder and the material parameters for each shaft section are configured. Second, the 4-DOFs radial suspension forces $f_{xA}, f_{xB}, f_{yA}$, and $f_{yB}$ are added at the bearing coordinate system's origin and a revolute motion around the z-axis at the generalized coordinate system's origin. Third, 4-DOFs are taken to control current $i_{xA}, i_{yA}, i_{xB}$, and $i_{yB}$ as input and 4-DOFs displacement $x_{sA}, y_{sA}, x_{sB}$, and $y_{sB}$ as output. The Ansys Twin Builder plant module is then developed to characterize the rotor system. Forth, the closed-loop rotor system is created, seen in Figure 13, by importing the Ansys Twin Builder plant module into Simulink and adding the suspension controller Gc(s), power amplifier Gp(s), and displacement sensor Gs(s).

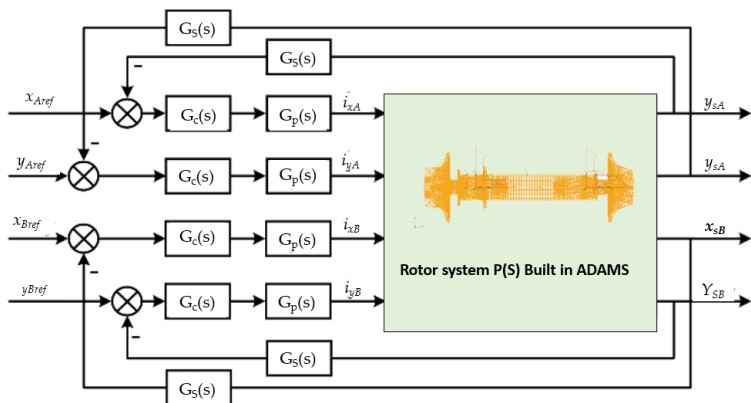

**Figure 12.** The physical model for the rotor's closed-loop system.

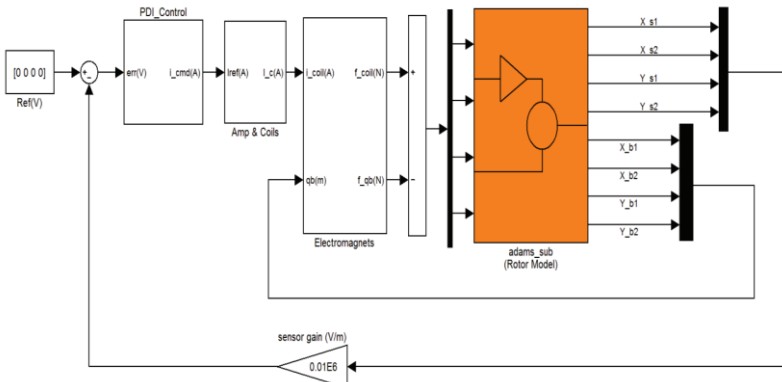

**Figure 13.** Simulation model system of high-speed motor for magnetic bearing.

In the MATLAB/Simulink software package, the control system and electromagnetic actuator models were created. Figure 11 illustrates the overall design. The input to the Ansys Twin Builder model is the four electromagnetic forces in the radial active magnetic bearings, and the output of the Ansys Twin Builder plant model is the displacements of the rotor at the sensors. The actuator model uses the rotor deviations on each radial bearing to compute the position stiffness forces, and the controller uses the rotor deviations on each sensor to create the control current. A flexible rotor suspended by radial AMBs is modeled in Ansys Twin Builder /View and imported into MATLAB/Simulink as a Ansys Twin Builder sub-module, which includes a function module. Figure 14 illustrates the Ansys Twin Builder sub-module contents.

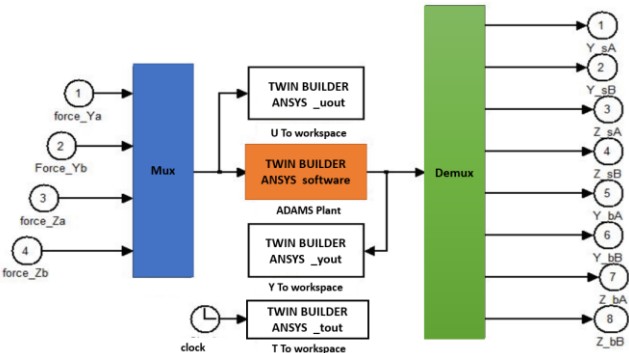

**Figure 14.** Mechanical system model of rotor.

The mass imbalance of the rotor is an essential result of factors such as unequal shaft material and mistakes made during machining and assembly, which leads to the geometric axis not aligning with the inertial axis. On the one hand, the static imbalance and vibration

force are caused by the offset between the geometric axis and the inertia axis. On the other hand, the dynamic imbalance and vibration torque are caused by the displacement between the geometric and inertia axes. One way to express the irregular movement between the geometric axis and the inertia axis is as follows:

$$q_\Delta = \text{qi} - \text{qg} \tag{20}$$

According to Equation (12), the following form can also be used to describe the eccentric movement.

$$q_\Delta = \begin{bmatrix} \sigma\cos(\Omega t + \gamma) \\ \xi\cos(\Omega t + \theta) \\ \sigma\sin(\Omega t + \gamma) \\ \xi\cos(\Omega t + \theta) \end{bmatrix} \tag{21}$$

where $\Omega$ and $\theta$ are the amplitude and phase of static imbalance, $\sigma$ and $\gamma$ are the amplitude and phase of dynamic imbalance. Taking into account the rotor mass imbalance, the rotor dynamics described by Equation (12) can be rewritten as

$$\text{m}(\ddot{q}_g - \ddot{q_\Delta}) + \text{G}(\dot{q}_g - \dot{q_\Delta}) = \text{T}_b\text{K}_x T_b^T \text{q}_g + \text{T}_b\text{K}_i\text{I} \tag{22}$$

Equation (20) shows that the vibration induced by rotor mass imbalance has the same frequency as the rotational speed as a result of the phase-shift drop filter. The quantitative link between recorded synchronous displacement and rotor mass imbalance can then be determined. The phase-shift notch filter is defined as

$$\text{N(s)} = \frac{s^2 + \Omega^2}{s^2 + k \times cos\Psi + \Omega^2 - k \times sin\Psi} \tag{23}$$

where k is the gain of the high pass filter, and $\Psi$ is the gain of the high pass filter and is the stable suspension compensation phase. With a high pass filter, the synchronous current is suppressed to zero, and the rotor dynamics in Equation (22) are simplified as

$$\text{m}(\ddot{q}_g - \ddot{q_\Delta}) + \text{G}(\dot{q}_g - \dot{q_\Delta}) = \text{T}_b\text{K}_x T_b^T \text{q}_g \tag{24}$$

Since a thin rotor has a weak rotating effect (G = 0), radial translation and rotation can be considered separately. The movement of the rotor's geometric center in the direction of translation is defined as g(t) and the movement in the direction of rotation as $v_g(t)$. The strange word is defined as $\eta\Delta(t)$, $v\Delta(t)$. They can be shown in particular ways as

$$\eta_g(t) = x(t) + jy(t)\eta_\Delta(t) = x_\Delta(t) + jy_\Delta(t)Vc(t) = \beta(t) + j\alpha(t)v_\Delta(t) = \beta_\Delta(t) + j\alpha_\Delta(t) \tag{25}$$

Substituting Equation (25) into Equation (24) to find the equation of rotor dynamic results in

$$ms^2\big[\eta_g(s) + \eta_\Delta(s)\big] = 2k_x\,\eta_g(t)\text{I}_r s^2\big[vg(s) + v_\Delta(s)\big] = 2k_x l_b^2 v_g(s) \tag{26}$$

The $I_r = I_x = I_y$ and the radial rotor are symmetrical. By transforming Equation (21) into complex form, it becomes

$$\eta_\Delta(t) = \xi e^{j(wt+\theta)} v_\Delta(t) = \sigma e^{j(wt+\gamma)} \tag{27}$$

Substituting Equation (27) into Equation (26) obtains

$$\xi e^{j(wt+\theta)} = -\frac{mw^2 + 2k_x}{mw^2} A_\eta\, e^{j(wt+\varnothing_\theta)} \sigma e^{j(wt+\gamma)} = -\frac{I_r w^2 + 2kx l_b^2}{I_r w^2} A_v\, e^{j(wt+\varnothing_\gamma)} \tag{28}$$

According to Equation (28), the amplitude and phase of the rotor mass imbalance can be obtained online by

$$\xi = A_\eta \frac{mw^2 + 2k_x}{mw^2} \sigma = A_v \frac{I_r w^2 + 2kxl_b^2}{I_r w^2} \theta = \varnothing_\theta + \pi \gamma = \varnothing_\gamma + \pi \tag{29}$$

Furthermore, to model 3D vibration dynamics with an unbalanced rotor mass, the amplitudes and phases solved by Equation (29) are the same as the A-end and B-end counter discs. Use $m_A$ and $\varnothing_A$ for the mass and phase at the A-end of the counter disc. $m_B$ and $\varnothing_B$ are used to represent the mass and phase at the B end of the counter disc. This can be said about the counterweight matrix:

$$\text{m}\Delta = \begin{Bmatrix} m_A r_a \cos(\varphi_A) \\ m_B r_B \cos(\varphi_B) \\ m_A r_a \sin(\varphi_A) \\ m_B r_B \sin(\varphi_B) \end{Bmatrix} \tag{30}$$

The generalized coordinate system's force moment must be zero for the principle of equivalence to apply and there are

$$\begin{aligned} m_{bx} w^2 r_b l_{cB} - m_{ax} w^2 r_a l_{cA} &= I_r w^2 \sigma \cos \gamma \\ m_{ax} w^2 r_a - m_{bx} w^2 r_b &= mw^2 \xi \cos \theta \\ m_{by} w^2 r_b l_{cB} - m_{ay} w^2 r_a l_{cA} &= I_r w^2 \sigma \cos \gamma \\ m_{axy} w^2 r_a - m_{by} w^2 r_b &= mw^2 \xi \cos \theta \end{aligned} \tag{31}$$

where the mass components are divided into radial and orthogonal directions by the letters $m_{ax}$, $m_{ay}$, $m_{bx}$, and $m_{by}$. The counterweight radius is given by $r_a$ and $r_b$ and $\omega$ is the rotor's rotating angular velocity. Finally, the discovered rotor mass imbalance's equivalent-counterweight amplitudes and phases can be solved as

$$\text{m}_A = \sqrt{m_{ax}{}^2 + m_{ay}{}^2} \tag{32}$$

$$\text{m}_B = \sqrt{m_{bx}{}^2 + m_{by}{}^2} \tag{33}$$

$$\varphi_A = \arctan(\frac{m_{ay}}{m_{ax}}) \tag{34}$$

$$\varphi_B = \arctan(\frac{m_{by}}{m_{bx}}) \tag{35}$$

The logical model is mainly used to establish dynamics consistent with the physical model. The logical model needs to be simultaneous simulation and remote control of digital twin models. In order to make the virtual department and accurately simulate the system's running process, it is necessary to convert various codes of the entity model, such as MATLAB apps program, etc. We can use information from the connected asset to define a model in MATLAB. Using its multi-domain modeling capabilities, Simulink may also be used to develop a model based on physical principles. In order to create a digital twin, data-driven or physics-based models can be fine-tuned with information from the live asset. These digital duplicates have many applications, including but not limited to prediction, what-if analysis, anomaly detection, fault isolation, and more.

MATLAB facilitates data-driven approaches such as machine learning, deep learning, neural networks, and system identification (45:55). A model is often trained or extracted from one set of data and then validated or tested on another. For identification and analysis, in this paper, a MATLAB app neural network is used for a data-driven approach.

The logical model deals with the virtual reality interaction mechanism, code analysis, simulation, and remote monitoring. According to the digital twin data transmission reference architecture, the primary input data transmission sources are the physical model. Corresponding to the data interface technology, the input–output flow is mainly designed with the RESTful architecture. The RESTful server is opened to monitor data requests by MATLAB and deal with two-way connection data requests/response from the magnetic bearing to MATLAB and vice-versa. According to the specific request/response, order data are input and output in HTTP-XML/JSON format. The requested system completes the transfer data (as shown in Figure 15).

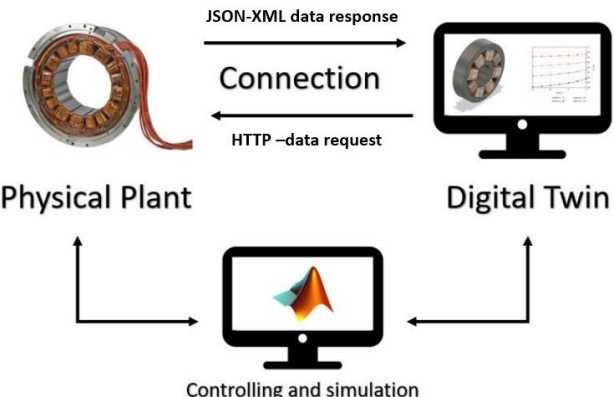

**Figure 15.** Flow processing of the transfer data MB.

According to the actual needs of the digital twin production magnetic bearing equipment monitoring system, mainly the output data of the physical layer are monitored. The running status data of the equipment data request can be divided into real-time data and historical data. Finally, we will explain the third model of the behavior layer, which is the data collection and the comparison study between the physical layer and the behavior layer in the results section.

## 3. Results

Related to the behavior layer, the data-collecting model involves the online identification of rotor mass imbalance and vibration dynamics modeling verification of magnetic bearings. At first, the feasibility of the suggested technique for identifying static mass imbalance is tested for the purpose of static mass imbalance identification and displacement control. In the identification simulation, two excitation signals were utilized, namely $F_{c1} = 15 \sin \omega t$ and $F_{c2} = 18 \sin \omega t$. The selection of the integral period is such that it is 120 times greater than the rotation period. Figures 16 and 17 present the Fourier coefficient of the sensor1 and sensor2. After completing the integral process, the Fourier coefficients of the displacement sensor signal with the excitation signals $F_{c1}$ and $F_{c2}$ are shown in Figures 17 and 18. These coefficients are $\alpha_{s1} = 0.459$, $\beta_{s1} = 0.254$, $\alpha_{s2} = 0.487$, $\beta_{s2} = 0.233$. Then at 5 s, the rotor mass imbalance online identification algorithm is presented.

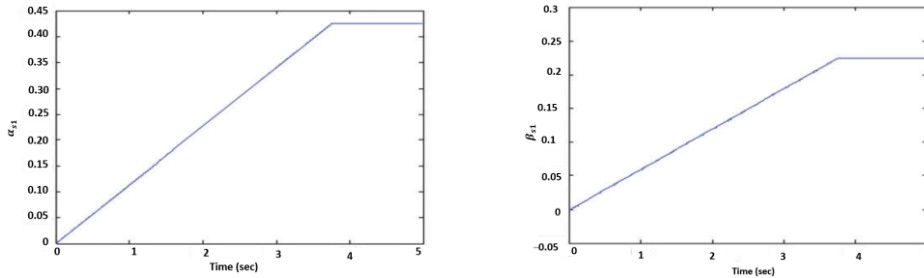

**Figure 16.** Fourier coefficients ($\alpha_{s1}, \beta_{s1}$) of displacement sensor1 signal.

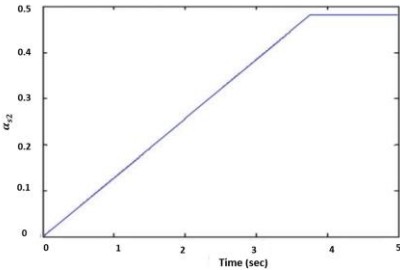 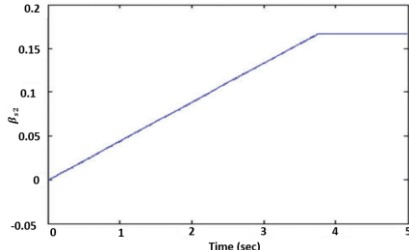

**Figure 17.** Fourier coefficients ($\alpha_{s2}$, $\beta_{s2}$) of displacement sensor1 signal.

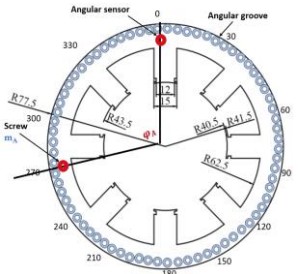 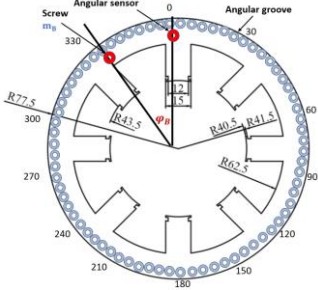

**Figure 18.** Equivalent counterweight of A-end and B-end schematic diagram.

The amplitude and phase of the counterweight disc at the A-end is 743.5898 mg and 270°, respectively, whereas those at the B-end are 258.75 mg and 335°, as calculated by Equations (31)–(35). The comparable counterweight on the double-ended counterweight disk is depicted in a schematic form in Figure 18. The vibration behavior model with rotor mass imbalance can be recreated by fixing mass blocks at various points on the 3D model of the counterweight discs.

Figure 18 presents the findings of the equivalent counterweight. It can be seen from Figure 19, showing the rotor displacement trajectory for the A-end and B-end, that the B-end has an equivalent double-ended counterweight higher than A-end, consistent by comparison with the measured displacement trajectory.

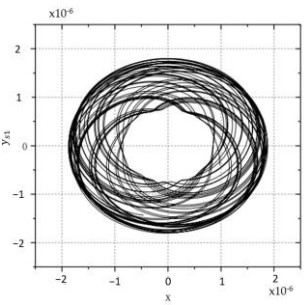 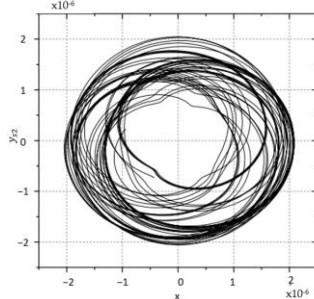

**Figure 19.** Output sensors for rotor displacement trajectory for A-end and B-end.

The displacement sensor signals, as depicted in Figure 20, are derived from the two excitation signals. As depicted in Figure 20, the synchronous component of the displacement sensor exhibits a significant magnitude. Figure 21 presents the current at the A-end and B-end magnetic bearings with respect to time.

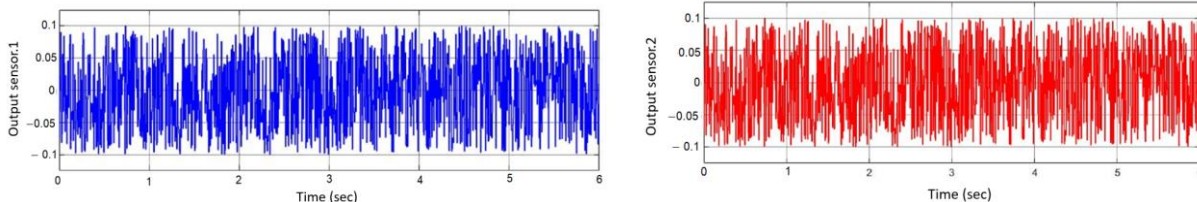

**Figure 20.** Output signal of the displacement sensor1 and sensor2.

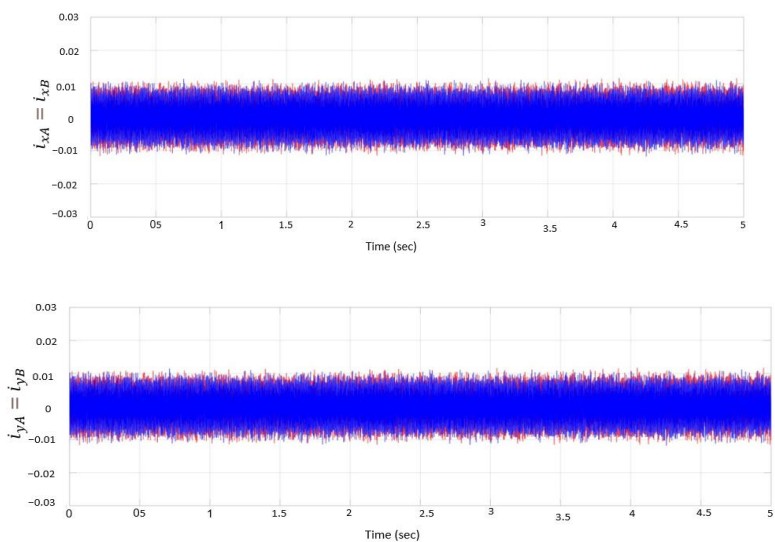

**Figure 21.** Current results at the magnetic bearing.

As can be observed below, the displacement trajectory of the rotor is quite similar to the measured trajectory when the corresponding double-ended counterbalance is used.

Figure 22 shows the effectiveness and accuracy of the proposed physics layer method for vibration acceleration dynamics with rotor mass imbalance, related to the A-end and B-end.

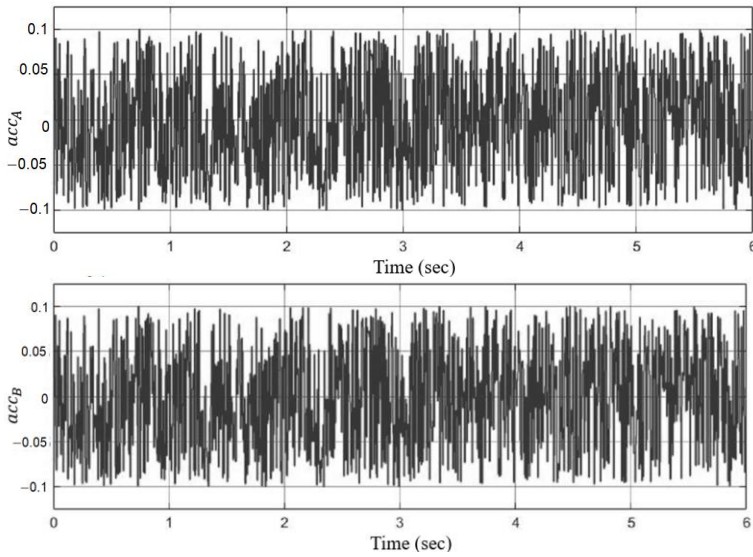

**Figure 22.** Vibration acceleration predicted by A-end and B-end disc.

Figure 22, contain the Vibration acceleration $acc_A$ is depicted, and $acc_B$ close to the A-end and B-end counterweight discs, respectively (outcome acceleration of physical model). Figure 23 present the online identification of acceleration on the A-end and B-end. Since

more mass is being added to the B-end of the disc, $acc_B$ has a larger amplitude than $acc_A$. The blue line represents the online identification data of the A-end, while the red line represents the online identification data of the B-end.

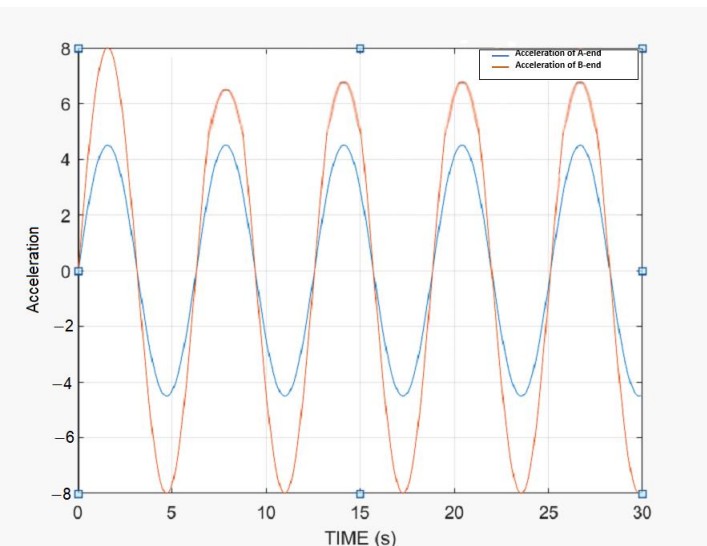

**Figure 23.** The vibration acceleration predicted by transmission online.

Finally, the vibration transmission with the verification of the digital twin system results are presented in Figure 23. From Figure 23, it can be concluded that the B-end disc imbalance is increased to test the efficiency of the vibration transmission network. The vibration transmission with mass imbalance from the physical layer is verified with the training samples of vibration dynamics in the behavior layer to confirm the work of the digital twin modelling network, as presented in Figure 24; when comparing the red and blue curves, it is clear that the digital twin is performing as intended because the predicted signal's amplitude and phase closely match those of the real signal.

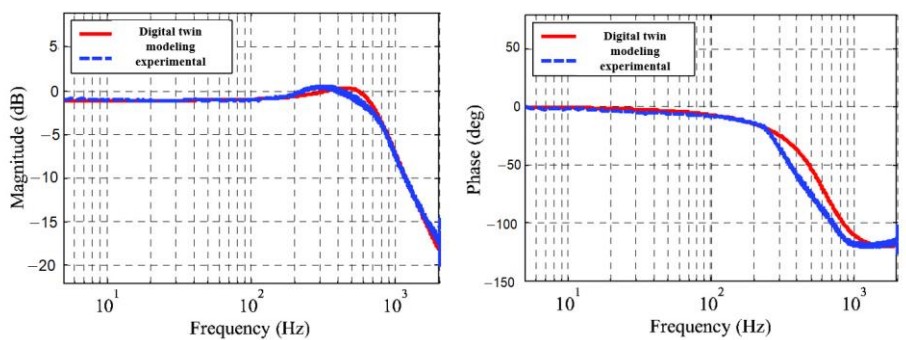

**Figure 24.** Comparison of the vibration transmission with mass imbalance.

## 4. Conclusions

In this paper, a digital twin model of a magnetic bearing with driven vibration is proposed. Results from simulation and testing illustrate the effectiveness of the online identification method's vibration transmission network, 3D vibration dynamics modeling, and rotor mass imbalance. In order to make simulations of vibration dynamics more realistic, researchers plan to investigate new ways of representing vibration sources that produce many frequencies. Independent factors include amplitude (300 mg, 700 mg), mass imbalance phase (0°, 90°, 180°, 270°), and rotor speed (30 Hz, 50 Hz). There are 200 epochs utilized in total, with a nine to one ratio between the training and test sets. The result is anticipated by the trained vibration transmission network with a rotor speed of 30 Hz

(1800 rpm), a 300 mg amplitude, and a 0° phase. The results present the comparison of phase and magnitude transmission networks. i.e., the difference between experimental work (physical model) and the behavior layer to confirm the work of the digital twin. The results show that using the digital twin in the magnetic bearing will make it more reliable, perform better, and failure can be predicted before it occurs. For future research, digital twins for magnetic bearings may be integrated with other systems, such as control systems, electronic power converters, or motor drives, to create a more comprehensive model of the overall system and optimize its performance.

**Author Contributions:** Conceptualization, O.W.T.; methodology, O.W.T.; software, O.W.T.; formal analysis, O.W.T.; investigation, O.W.T.; writing—original draft preparation, O.W.T.; writing—review and editing, O.W.T.; visualization, Y.H.; supervision, Y.H.; funding acquisition, Y.H. All authors have read and agreed to the published version of the manuscript.

**Funding:** This research was funded by the National Key Research and Development Program of China: 2018YFB2000103.

**Institutional Review Board Statement:** Not applicable.

**Informed Consent Statement:** Not applicable.

**Data Availability Statement:** Not applicable.

**Acknowledgments:** To my family, especially my great wife (Issraa Jwad Kazim), without whom I would not have been able to write this paper.

**Conflicts of Interest:** The authors declare no conflict of interest.

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
