# Peer review of "Modeling of a Digital Twin for Magnetic Bearings"

_applsci, doi:10.3390/app13148534_

Round 1

Reviewer 1 Report

The key contribution of the paper seems to be the model of magnetic bearing but not in the term of digital twin.

Major concern

1. There are three levels of digital twin: Digital Model, Digital Shadow, and Digital Twin, depending on how the data is exchanged between the physical and digital objects. It appears that the paper does not clearly illustrate the relationship between the physical and digital objects. The paper shows that if the inputs of the physical and digital objects are the same, the results will also be the same. This interpretation contradicts the definition and essence of a digital twin.

2. The paper has been check with the turnitin website.  It seems that the paragraph  about Digital Twin is generated by AI. So not sure if the authors miss understand the digital twin concept. The digital twin is categorized by product lifecycle phases, design, manufacturing, service, retire, and full lifecycle phase. The way that present in the paper is very limited. The paper should focus on how to develop the dynamic model of magnetic bearing,  not focus on digital twin.

3. In section 2.2 of the paper, the authors fail to adequately describe the model, which is a key aspect of their research. The section begins with the sentence, "The upper electromagnets in the y direction (vertical) obtained 5 A," which lacks clarity and does not effectively convey the intended message. To improve the paper, the authors should address the following topics:

  1. System Description: The authors should provide a detailed description of the system and its components. Additionally, including photographs of the system would be beneficial for better understanding.

  2. Free Body Diagram: It is important to present the free body diagram of the system and its components. This diagram will help readers visualize the forces acting on the system. Furthermore, a system diagram illustrating the relationships between the components should be included.

  3. Derivation of Dynamic Equations: A step-by-step explanation of how the dynamic equations are derived is necessary. The authors should clearly outline the process and highlight any differences between the dynamic equations proposed in their paper and those presented in previous research.

  4. Control Technique: The authors should describe the control technique employed in their study. It is crucial to explain the unique aspects or innovations of their approach in comparison to existing control techniques described in the literature.

  5. Simulation Results and Verification: The paper should include the simulation results obtained from the proposed model. The authors should explain how they verified these results, ensuring transparency and reliability.

Improving the clarity and inclusion of these aspects will enhance the quality and comprehensibility of the paper.

4. The authors fail to define a lot of parameters in equations, figures, and paragraphs. for example, when the reader read the line 132 about Fig.1. The reader will understand what I0 Icy Ics are. Please define all parameters and define them step by step.

5. What is the difference between Figure 1 and 5?

6. Why the number of magnetic cores in Figure 6 does not match to the number of magnetic cores in other figures such as Figure 1, 4, 5and 7?

7. What is Figure 7? How is it relate to the digital twin? It is look like only pages of power point? What is the data flow?

8. What is Figure 8? How is it relate to the Figure 1 or Figure 5?

9. The quality of the most figure is not good. For example , Figure 1, 2, 3, 5, 6, 7, 17.

10. There are a lot of meaningless figures in the result sections. For example Figure 19, 20, 21.

11. No explanation of physical system.

12. No explanation how to set the experiment and condition.

13. Still have a lot of comments and a lot of questions in the paper.

Author Response

Thank you for your given attention to carefully reviewing our manuscript in your busy schedule. You can see the reply to comments in the attachment file.

Reviewer 2 Report

Quality of English need to be improved, I have also commented. 

Author Response

(The authors gave the same response as above.)

Reviewer 3 Report

The authors studied the performance of the magnetic bearing system simulated in real-time using a digital twin, especially the resulting vibration from the unbalanced rotor mass, which caused a drop in performance and a high risk of system instability and potential safety accidents. My constructive criticisms of the article are itemized below:

1.     The abstract should be revised to reflect the motivation for the study, the methods utilized, results, and conclusions drawn.

2.     Please state your motivation, goals and objectives, the novelty of this research, and potential contribution to the literature. Mention these in the last paragraph of the Introduction.

3.     In its current form, the contribution of the paper is not so clear. Do the authors have the ambition to develop a fundamentally new approach or do they simply apply existing methods to a new test case, in which they compare experimental data to numerical results?

4.     How did the authors determine the material parameters used in Table 1?

5.     How did the authors define the material parameters used in Tables 2 and 3?

6.     Too many mathematical equations were presented without proper referencing to the piezoelectric composite configuration and/or physics compromised the impact of this paper on the readers in the smart materials and structures field.

7.     In practical applications, how do the authors select the best position for the sensor, since if the wrong position is chosen, the results are not satisfactory?

8.     How do the authors treat the aliasing and leakage phenomena in dynamic analysis?

9.     An important issue that should be better addressed by the authors is regarding the analysis threshold, the influence of the manufacturing process, and process uncertainties.

10.  What are the application fields of the proposed methodology, in terms of their impact on the overall efficiency of the applied system?

11.  The limitations must be discussed by the authors and to the related need of further work.

12.  I’m not an English language specialist, but many typing and grammar errors could make the reader confused. I suggest reviewing the entire manuscript.

Short sentences and out of context. It must be improved.

Author Response

(The authors gave the same response as above.)

Round 2

Reviewer 1 Report

1. No photos of physical plant

2. No explanation what the flow data between digital and physical plant

3. No experiment on digital twin. There are only simulations on math model.

4. No comparison results between math model and physical plant.

5. Reader cannot simulate the results in the paper. Please provide more details in simulation step by step.

6. Which equations do you to simulate fig. 20.

7. The authors fail to present the digital twin concepts. No implementation. Only simulation.

8. Fig 6 should be the key of digital twin. It should show the connection and data flow and should not be some power point slide.

9. The authors use 8 pole of bearing in math model and simulation but slide in fig 6 has 12 pole. Why?

11. How do the authors know that the results are corrected?

12. There are a lot more comments. 

Author Response

(The authors gave the same response as above.)

Reviewer 2 Report

Author have addressed the comments, now it it feasible for possible publication. 

Much improved now 

Author Response

Thank you for your given attention to carefully reviewing our manuscript in your busy schedule. 

Reviewer 3 Report

Although the authors answered the reviewers' questions, such modifications are not highlighted in the work. Therefore, I advise authors to highlight changes in the article for a better evaluation.

Although the authors answered the reviewers' questions, such modifications are not highlighted in the work. Therefore, I advise authors to highlight changes in the article for a better evaluation.

Author Response

Thank you for your given attention to carefully reviewing our manuscript in your busy schedule. We improve the manuscript.